# Adjuvant Use of Pembrolizumab for Stage III Melanoma in a Real-World Setting in Europe

**DOI:** 10.3390/cancers16213558

**Published:** 2024-10-22

**Authors:** Michael Weichenthal, Joanna Mangana, Iva Gavrilova, Iwona Lugowska, Gergana Krumova Shalamanova, Lidija Kandolf, Vanna Chiarion-Sileni, Peter Mohr, Teodora Sotirova Karanikolova, Pawel Teterycz, Enrique Espinosa, Philipp Schnecko, Phil Cheng, Marc Bender, Shan Jiang, Thomas Burke, Paolo Antonio Ascierto, Helen Gogas, Ivan Marquez Rodas, Piotr Rutkowski, Dirk Schadendorf, Reinhard Dummer

**Affiliations:** 1Skin Cancer Center Kiel, University Hospital Schleswig-Holstein, 24105 Kiel, Germany; 2University Hospital of Zurich, 8091 Zurich, Switzerland; johanna.mangana@usz.ch (J.M.); phil.cheng@usz.ch (P.C.); reinhard.dummer@usz.ch (R.D.); 3National Oncology Hospital Sofia, g.k. Darvenitsa, 1756 Sofia, Bulgaria; ivaga81@yahoo.com; 4Instytut im. Marii Skłodowskiej-Curie, 02-781 Warsaw, Poland; iwona.lugowska@pib-nio.pl (I.L.); pawel.teterycz@pib-nio.pl (P.T.); piotr.rutkowski@pib-nio.pl (P.R.); 5Complex Oncology Center Plovdiv, 4004 Plovdiv, Bulgaria; gery_dd@abv.bg; 6Military Medical Academy (CSEEMEG), 11000 Belgrade, Serbia; lkandolfsekulovic@gmail.com; 7Istituto Oncologico Veneto, IOV-IRCCS, 35128 Padova, Italy; vanna.chiarion1@gmail.com; 8Elbe Kliniken Buxtehude-Stade, 21614 Buxtehude, Germany; peter.mohr@elbekliniken.de; 9Nadezhda Hospital, 1330 Sofia, Bulgaria; teddy.karanikolova@abv.bg; 10Hospital Universitario La Paz, 28046 Madrid, Spain; e.espinosa@salud.madrid.org; 11Alcedis GmbH, 35394 Gießen, Germany; 12Skin Cancer Center, Division of Molecular Cell Biology, Elbe Kliniken Stade-Buxtehude, 21614 Buxtehude, Germany; marc.bender@elbekliniken.de; 13Merck & Co., Inc., Rahway, NJ 07065, USA; shan.jiang1@merck.com (S.J.); thomas_burke2@merck.com (T.B.); 14Melanoma and Cancer Immunotherapy and Development Therapeutics Unit, Istituto Nazionale Tumori IRCCS Fondazione “G. Pascale”, 80131 Naples, Italy; paolo.ascierto@gmail.com; 15First Department of Internal Medicine, Laikon General Hospital, School of Medicine, University of Athens, 15772 Athens, Greece; helgogas@gmail.com; 16Hospital General Universitario Gregorio Marañón, 28007 Madrid, Spain; ivanpantic@hotmail.com; 17Universitätsklinikum, 45147 Essen, Germany; dirk.schadendorf@uk-essen.de

**Keywords:** melanoma, cutaneous melanoma, stage III, adjuvant pembrolizumab, EUMelaReg, real-world

## Abstract

We studied real-world patient trends and cancer survival in adult patients in a European registry, EUMelaReg. We included 200 patients with stage III melanoma with lymph node involvement who had complete resection and received adjuvant treatment with pembrolizumab. Patients initiated treatment with adjuvant pembrolizumab from 1 January 2019 to 17 April 2021 with a median follow-up of 16.5 months. Comparison with previously published real-world data showed that patients were older and more likely to have stage IIIC and IIID disease than those in the Keynote 054 clinical trial.

## 1. Introduction

Cutaneous melanoma is among the most frequently diagnosed cancer types in the majority of countries with fair-skinned populations, including Canada, the United States (US), Europe and Australia. In Europe, skin melanoma has an age-standardized incidence of 8.1 to 14.6 per 100,000 individuals [1]. While most melanomas are cured by simple excision, metastasis occurs in approximately 15% to 40% of cases, and despite recent improvements in systemic therapy, leads to death in at least 50% of these cases [2,3,4].

In the past 15 years, enormous progress has been made in the prevention and treatment of metastatic melanoma due to the development of BRAF and MEK inhibitors for BRAF V600-mutated melanoma, and even more importantly, immune checkpoint inhibitors (ICIs) [4,5]. Following on the successful results of trials in advanced metastatic melanoma, ICI therapies also showed favorable results in the adjuvant treatment of completely resected locoregional disease [6,7]. Anti-PD1 antibodies were subsequently approved in Europe for the adjuvant treatment of melanoma with completely resected locoregional disease, and in December 2018, the European Medicine Agency (EMA) approved pembrolizumab for the adjuvant treatment of adults with stage III melanoma.

The prospective randomized phase III Keynote 054 (KN054) study evaluated 12 months of adjuvant pembrolizumab therapy in stage III completely resected melanoma patients [7,8,9]. At a median follow-up (FU) of 15 months, recurrence-free survival (RFS) was significantly longer in the pembrolizumab group than in the placebo group (hazard ratio [HR] 0.57; 98.4% confidence interval [CI] 0.43–0.74, *p* < 0.001) [7] and after an overall median FU of 42.3 months, compared to placebo, pembrolizumab adjuvant therapy significantly improved distant metastasis-free survival (DMFS) (HR 0.60; 95% CI 0.49–0.73, *p* < 0.0001). Despite these improvements in the treatment of metastatic melanoma, there are substantial differences in survival rates as well as access to innovative treatments for melanoma among European countries, suggesting the existence of significant inequalities in healthcare [10,11].

Although data on patients treated with adjuvant pembrolizumab are available from clinical trials and in the real-world in single countries, there are no multi-country real-world data studies investigating the use of pembrolizumab as adjuvant treatment for stage III melanoma patients. Therefore, this retrospective study was performed to analyze the treatment pattern and clinical outcomes of resected stage III melanoma patients who were treated with adjuvant pembrolizumab on European level using data from the European Melanoma Registry (EUMelaReg). EUMelaReg was founded to address the real-world treatment of melanoma and the outcome of patients across Europe and Israel [12]. It is a disease-entity-based treatment registry specific to collect real-world data on the available diagnoses and treatment patterns of melanoma patients at the European level.

## 2. Materials and Methods

### 2.1. Study Design and Patient Selection

This retrospective observational study analyzed adult (age ≥ 18 years) patients from the EUMelaReg database who were diagnosed with resected stage III cutaneous, or melanoma of unknown primary (MUP) and received at least one administration of adjuvant pembrolizumab between 1 January 2019 and 17 April 2021. The country registries that contributed patient-level data to this study were Bulgaria, Croatia Serbia, Bosnia and Herzegovina, Germany, Greece, Italy, Poland, Spain and Switzerland.

Patients were treatment naive for any anti-cancer drugs and must have had at least 12 months of FU and a survival status after the first administration of adjuvant pembrolizumab. Patients were excluded if they had uveal melanoma or received pembrolizumab therapy within a clinical trial or expanded access program.

### 2.2. Outcomes

The primary objectives were to describe the demographic and clinical characteristics and treatment history, time on adjuvant pembrolizumab treatment (TOT), RFS and DMFS from initiation of pembrolizumab. Secondary objectives included time to next treatment (TTNT) after adjuvant pembrolizumab treatment, next-line therapy and OS from the start of adjuvant pembrolizumab.

### 2.3. CT/MRI

All patients were staged according to AJCC classification 8th edition. Depending on the country, organs including the brain were screened for metastasis using different techniques and at different frequencies. Computed tomography (CT) or magnetic resonance imaging (MRI) or positron-emission tomography (PET) or PET-CT scans were performed mostly every 3 to 6 months during adjuvant treatment and every 6 months post treatment for up to 3 to 5 years in most of the countries.

### 2.4. Statistical Analysis

Descriptive statistics were used to summarize baseline study cohort characteristics. For categorical variables, the data are presented as the number of observations and the percentage. For continuous variables, data are presented as the mean ± standard deviation (SD), median, minimum and maximum. For TOT, RFS, DMFS, TTNT and OS, time-to-event analyses were conducted using the Kaplan–Meier method to generate Kaplan–Meier plots and to estimate median time-to-event in months with 95% CI, and events rates with 95% CI at landmark timepoints. TOT is defined as the time from the date of start of treatment to the date of end of treatment. A patient’s date of end of treatment was used as a cutoff date for that patient regardless of whether the treatment was documented as ongoing. RFS is defined as time from the start date of adjuvant pembrolizumab treatment to the date of the first recurrence according to the physician’s assessment or death due to any cause, whichever occurred first. Patients were censored at the start of the next treatment. If neither a subsequent treatment nor death was documented, a patient was censored with the date of last contact. DMFS was defined as time from the start date of adjuvant pembrolizumab treatment to the date of the first documentation of distant metastasis according to the physician’s assessment or death due to any cause, whichever occurred first. Patients were censored at the start of the next treatment. If neither a subsequent treatment nor death was documented, a patient was censored with the date of last contact. TTNT under real-world conditions was calculated for all patients from the start of the adjuvant pembrolizumab therapy to the start date of the next treatment or death, whichever occurred first. All other patients were censored at the last date they were known to be alive. OS is defined as the time from the start date of first pembrolizumab treatment to the date of death due to any cause. The OS for subjects not known to have died was censored at the last date the patient was known to be alive. Follow-up was calculated by Kaplan–Meier analysis with last contact as event and fatal events censored.

For stratified (i.e., sub-group) analyses, Kaplan–Meier plots were created for time to event analyses. Single variable Cox proportional hazards models were analyzed with the stratification factor as the independent variable subject to sample size considerations (a minimum of 10 patients in each level of the stratification factor). Adjacent categories could be combined to meet the minimum number of patients criteria. The HR and 95% CI were summarized for each level of the stratification factor.

All descriptive statistical analyses were performed using SAS statistical software (version 9.4 or higher). For survival analyses, the R packages *survival* and *survminer* were used. The patients were stratified for the following factors, which included more than 10 patients in each group: sex, age, BRAF status and American Joint Committee on Cancer (AJCC) stage (except stage IIID).

## 3. Results

### 3.1. Baseline Characteristics

A total of 200 eligible patients were extracted from the EUMelaReg database for this study. Demographic and patient characteristics at the time of adjuvant pembrolizumab treatment are summarized in Table 1. In total, 117 (58.5%) male patients and 83 (41.5%) female patients were treated with adjuvant pembrolizumab after complete resection. The overall median age was 63 (19.0–88.0) years. Male patients were older (median age: 64.0 [22.0–85.09] years), had a slightly higher percentage of diagnosed MUP of 6.8% and a lower BRAF mutation status (33.3%) compared to female patients with a median age of 59.0 (19.0–88.0) years, 2.4% MUP and 42.2% BRAF mutation (Table 1 and Appendix A).

The most prevalent stage according to the AJCC 8th edition criteria was stage IIIC with 60.0%, followed by stage IIIB with 26.0%, stage IIIA with 10.5% and stage IIID with 3.5% of the total population. The proportion of stage IIIC/D was higher in male patients (66.7%), older patients (≥70 years: 67.2%) and BRAF-negative patients (70.6%) compared to female patients (59.0%), younger patients (>70 years: 61.8%) and BRAF-positive patients (54.1%) (Table 1 and Appendix A). Stratification of patients by AJCC stage showed the highest proportion of stage IIIC/D in male, older and BRAF-negative patients (Appendix A).

In total, 78% of the total population had an Eastern Cooperative Oncology Group (ECOG) performance status of 0, which correlated with age, BRAF status and AJCC staging. The higher the age and staging of the patients, the lower the proportion of ECOG status 0 (Table 1 and Appendix A). In total, 47.0% of the patients presented with at least one documented comorbidity.

### 3.2. Survival Analyses

#### 3.2.1. Time on Adjuvant Treatment

Time on treatment (TOT) with pembrolizumab and reason for discontinuation stratified by gender are shown in Table 2. Median TOT (95% CI) was 11.1 (9.2–11.5) months in the total population (Table 2 and Figure 1A) and longer in female patients (11.1 [9.2–11.5] months) than in male patients (9.9 [6.9–11.6] months). The highest TOT probability (95% CI) was 6 months with 66.5% (60.2–73.6), followed by 9 months with 58.3% (51.7–65.8) and 12 months survival with 48.7% (42.0–56.6).

Stratification of TOT by 6, 9 and 12 months showed that the probability of TOT differ between gender, age, BRAF status and melanoma stage. Patients who were younger (<70 years) and female had a higher chance to stay on treatment for 12 months compared to older (≥70 years) and male patients (Table 3 and Appendix A). TOT (95% CI) at 12 months was 52.2% (44.1–61.8) vs. 41.1% (30.0–56.2) for patients >70 years and ≥70 years and 52.2% (44.1–61.8) vs. 41.1% (30.0–56.2) for female and male patients, respectively (Table 3 and Appendix A).

In total, 49.9% (95% CI: 40.5–61.4) of BRAF-negative and 41.8% (95% CI: 31.7–55.0) of BRAF-positive patients stay on treatment for 12 months, but the mutation status had no effect on the time and discontinuation of adjuvant treatment at earlier time points (TOT at 6 months: 64.0%; 95% CI: 55.0–74.5 vs. 64.9%; 95% CI: 54.9–76.7).

In addition to age, gender and BRAF status, the melanoma stage has an impact on the discontinuation of adjuvant treatment. Patients with stage IIIA had a 6-month TOT (95% CI) of 85.7% (72–100) and 12-month TOT of 61.5% (42.5–89.2) compared to patients with stage IIIC with 6-month TOT of 63.1% (37.9–72.6) and 12-month TOT of 43.1% (34.7–53.6) (Table 2 and Figure 2). However, male patients, older patients (≥70 years) and patients with stage IIIC/D discontinued treatment more frequently at earlier time points (Table 3 and Figure 2).

The most common reasons for stopping adjuvant treatment were regular completion of treatment (n = 72, 36.0%) and disease recurrence (n = 53, 26.5%). In total, 10% (n = 20) of patients discontinued adjuvant treatment due to tolerability. No differences were observed between female and male patients (Table 2).

#### 3.2.2. Recurrence-Free Survival and Distant Metastasis-Free Survival

Kaplan–Meier estimates show a median RFS of 29.6 (95% CI: 18.7–not reached [NR]) months and an 18-month RFS median rate of 59.6% (95% CI: 52.3–67.9) for the total population (Figure 1B). Age and gender had no effect on 18-month RFS. However, recurrence was correlated with melanoma stage, showing a higher RFS rate (95% CI) at 18 months of 76.6% (55.8–100) for stage IIIA, 66.7% (53.9–82.6) for stage IIIB and 53.0% (43.8–64.2) for stage IIIC (Table 4). The same trend was observed at 24 months, with a decrease in RFS with progression of AJCC stage (Appendix A). BRAF-negative patients had a slightly longer RFS rate (95% CI) at 18 months (59.5% (48.8–73.6) and 24 months compared to BRAF-mutated patients (18 months: 53.4 [42.4–67.2]) (Table 4 and Appendix A).

Median DMFS was 32.4 (95% CI: 22.7–NR) months and the median rate at 18 months was 70.0% (95% CI: 62.9–77.8) in the total population (Figure 1C). Subgroup analysis of DMFS showed a longer DMFS rate (95% CI) at 18 months for younger (72.2% [63.8–81.7]) and BRAF-negative (74.0% [64.5–84.9]) patients than for older (64.7% [52.5–79.9]) and BRAF-positive (61.1% [50.0–74.6]) patients. The same was observed at 24 months (Table 4; Appendix A). Stage IIIC patients exhibited distant metastases faster than stage IIIB and stage IIIC patients (stage IIIC: 64.4% [95% CI: 55.1–75.2]; stage IIIB: 77.2% [95% CI: 64.8–91.9]; stage IIIA: 82.0% [95% CI: 65.2–100]) at 18 months (Table 4 and Appendix A). However, median values were not significant (*p* = 0.32). DMFS appeared similar between female and male patients (Appendix A).

#### 3.2.3. Time to Next Treatment

Median TTNT was 29.9 (95% CI: 22.2-NR) months and the median 18-month survival rate (95% CI) was 64.5% (57.3–72.6) (Figure 1D and Table 5). At 18 months, TTNT (95% CI) was higher in BRAF-negative (66.2% [56.2–78.0]) than in BRAF-positive (54.9% [44.0–68.6]) patients, but similar between male and female patients and between elderly and younger patients (Table 5). At 24 months, there was no difference in TTNT between elderly and younger patients (Appendix A), male and female patients (Appendix A), or BRAF-positive and BRAF-negative patients (Appendix A). However, at all timepoints, patients with more advanced stages predictably had shorter TTNT, and at 24 months, TTNT (95% CI) was substantially better in stage IIIA (80.7% [62.5–100]) than in stage IIIB (59.4% [44.4–79.4]) and stage IIIC (46.3% [36.0–59.6]) patients (Appendix A).

#### 3.2.4. Overall Survival

Median 18-month OS rate (95% CI) was 88.1% (82.7–93.8) (Table 5) and lower in elderly (82.3% [71.5–94.8]) than in younger (90.6% [84.8–96.9]) patients, in BRAF-positive (81.0% [70.7–92.9]) than in BRAF-negative (91.4% [85.3–97.8]) patients and in stage IIIC patients (85.5% [78.2–93.4]) than in stage IIIB (91.2% [82.1–100]) and IIIA (100% [100–100]) patients (Table 5).

At 24 months, both male patients and elderly patients tended to have worse outcomes than female and younger patients (Appendix A). OS at 24 months was 100% for stage IIIA but was approximately 80% for stages IIIB and stage IIIC (Appendix A).

Previously observed differences can also be observed for OS survival. Differences in OS Kaplan–Meier curves show a better trend for younger (<70 years) patients, female patients and BRAF negative patient (Figure 3A–C). However, differences between female and male patients and between BRAF positive and negative patients are observed after 15 months. Stratification of OS by melanoma stage show the highest OS probability for stage IIIA patients. Patients with stage IIIB have a slightly better OS survival up to 18 months than patients in stage IIIC/D (Figure 3D).

## 4. Discussion

This retrospective study analyzed 200 fully resected stage III melanoma patients who were treated with pembrolizumab as an adjuvant therapy with a minimum of 1 year follow-up (median 16.5 months). This study provides real-world data obtained in a European setting, and therefore supplements other data on adjuvant anti-PD1 treatment of melanoma including the results of KN054 clinical trial, but without the usual restrictions of a clinical trial.

While randomized clinical trials remain the gold standard for evaluating the efficacy and safety of new cancer therapies, previous systematic reviews have demonstrated that clinical trials results may differ in important ways from those achieved in a real-world setting. Due to enrollment procedures and inclusion or exclusion criteria, a clinical trial may, for example, underrepresent certain demographic populations or miss differences in outcomes among disease status groups or subpopulations [13].

Real-world populations may therefore provide a less selective view with respect to demographic characteristics, e.g., in terms of age structure, performance status, comorbidities and other factors [13,14,15]. Hence, the current study was performed to evaluate the patient characteristics and survival outcomes of adjuvant pembrolizumab in a patient population in a real-world setting in Europe.

In KN054 only 8% of the patients were in IIIA as compared to 10.5% in EUMelaReg; IIIB 34.5 vs. 26.0, IIIC 49.7% vs. 60% and IIID 3.7% vs. 3.5%. From the AJCC 8th edition it is evidenced that stage IIIC is associated with a 5-year survival rate of 69% and 10-year survival rate of 60%, which correlates with EUMelaReg results and reflects the real-world situation.

Results may be also compared to a separate real-world study conducted using the US-based USON registry [16]. Treatment adherence was similar between the two real-world analyses, with a median duration of treatment at 11.1 (9.2–11.5) months in EUMelaReg patients and 11.8 (11.6–11.8) months in USON patients showing good therapy adherence in both cohorts. All three studies were performed in adult patients aged 18 years or older with stage III melanoma treated with complete resection and subsequent adjuvant pembrolizumab therapy.

We found that 12-month rates for RFS and DMFS in our study were substantially lower compared to KN054. These differences likely reflect differences in patient characteristics, as patients in our study were older (median age, 63 years) than those in the KN054 trial (median age, 54 years) and tended to have more advanced-stage melanoma (patients with grade IIIA-D accounted for 10.5%, 26%, 60% and 3.5% in EUMelaReg and 8.2%, 31.7%, 51.9% and 3.9% in KN054 with 4.3% unevaluable).

Nevertheless, comparing 12-month RFS rates for EUMelaReg and KN054 stratified by AJCC 8 stage still shows differences, which were especially pronounced in stage IIIC (60.1% in EUMelaReg vs. 73.6% in KN054). This might partially still be related to different substage distribution, since, e.g., patients with in-transit metastases were excluded in KN054, as were stage IIIA patients with a sentinel node tumor burden of less than 1.0 mm.

Conversely, in the USON real-world study RFS was higher (81.0%) than in both the KN054 or a real-world European setting. This could well be due to a shorter median follow-up time (9.3 months in the USON registry vs. 16.5 months in the EUMelaReg analysis and 15 months in KN054) and the substantially lower proportion of US registry patients who had more advanced-stage melanoma (only 40.4% of patients had stage IIIC and stage IIID melanoma). OS was not reached in any of the three studies.

Overall, we found that patient characteristics and risk profiles varied in important ways, including age and cancer stage, across the two registry-based real-world studies and the KN054 clinical trial. These differences are likely to have resulted in the numerically lower survival outcomes observed in our European study and the numerically higher survival outcomes observed in the US study when compared to the pivotal KN054 clinical trial.

This study has certain limitation related to the observational nature of the data collection leading to several sources of bias, including most importantly selection bias, missing data and underreporting of informative variables, such as comorbidities and other co-variates related to treatment selection or outcomes. Conversely, our study covered a more representative population than would normally be found in a clinical trial as our patients were not excluded based on ECOG score, age, co-morbidities or lymph node status. Additionally, the proportion of 37% of BRAF V600 mutated patients treated with pembrolizumab underscores the relevance of adjuvant immunotherapy in melanoma despite a given alternative of using BRAF/MEK-inhibitors in the adjuvant setting [17].

## 5. Conclusions

The patients in the EUMelaReg study had a lower recurrence-free survival outcomes than the clinical trial patients both in the overall population and by substage. This was likely due to a different spectrum of patients, i.e., the real-world patients seem to bear a worse prognosis, e.g., due to age and tumor substage. The study suggests that patient populations in clinical trials may not be fully representative for real-world populations, and therefore outcomes in clinical practice are important to study.

## Figures and Tables

**Figure 1 cancers-16-03558-f001:**
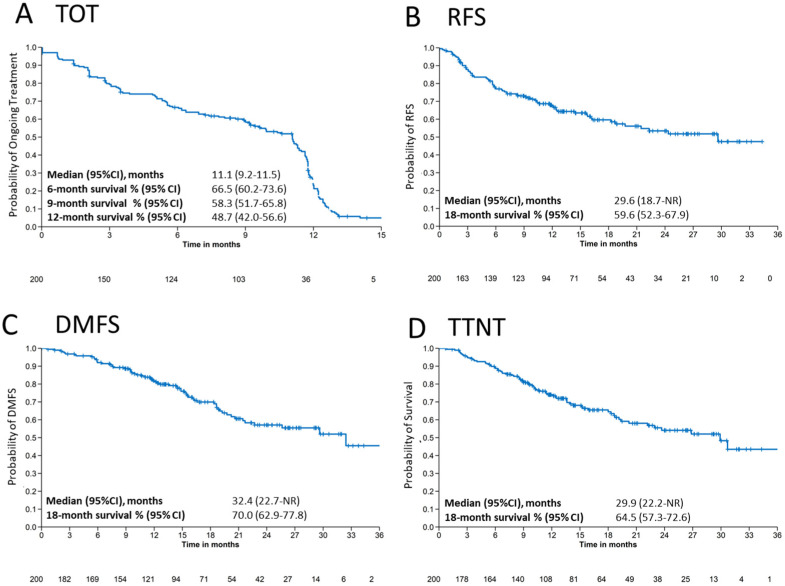
Survival outcomes. Kaplan–Meier estimates for (**A**) time on treatment (TOT), (**B**) recurrence-free survival (RFS), (**C**) distant metastasis free survival (DMFS) and (**D**) time to next treatment (TTNT) for patients treated with adjuvant pembrolizumab. CI: confidence interval, NR: not reached.

**Figure 2 cancers-16-03558-f002:**
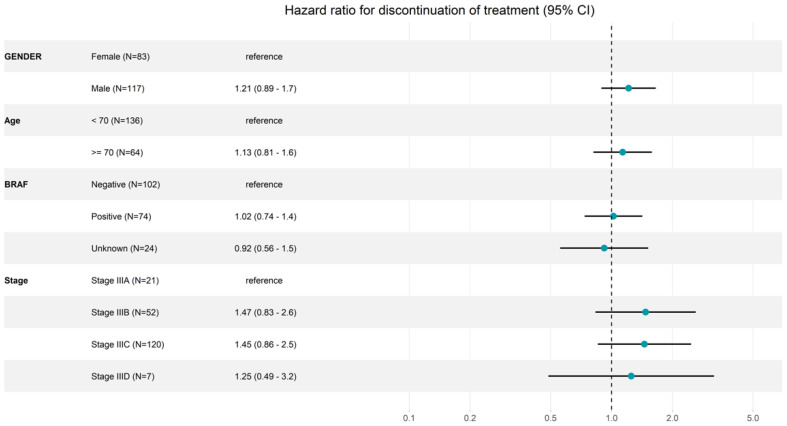
Forest plot for discontinuation of adjuvant pembrolizumab treatment. N: number of patients included in the analysis; Age: age in categories at therapy start; stage: AJCC (8th edition) stage at therapy start; CI: confidence interval.

**Figure 3 cancers-16-03558-f003:**
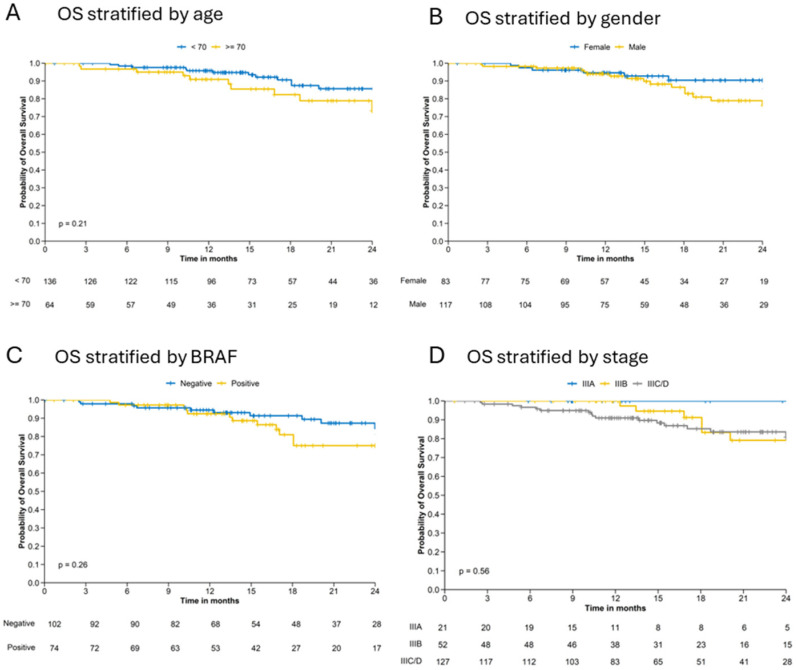
Kaplan–Meier estimates for overall survival (OS) stratified by (**A**) age, (**B**) gender, (**C**) BRAF mutation status and (**D**) stage.

**Table 1 cancers-16-03558-t001:** Patient demographics and disease characteristics at initiation of pembrolizumab treatment.

	Female (N = 83)	Male (N = 117)	Overall (N = 200)
**Age (years)**			
Mean (SD)	57.9 (15.3)	62.0 (14.5)	60.3 (15.0)
Median [Min, Max]	59.0 [19.0, 88.0]	64.0 [22.0, 85.0]	63.0 [19.0, 88.0]
**Age (years)**			
<70	61 (73.5%)	75 (64.1%)	136 (68.0%)
≥70	22 (26.5%)	42 (35.9%)	64 (32.0%)
**Melanoma subtype**			
Cutaneous melanoma	81 (97.6%)	109 (93.2%)	190 (95.0%)
MUP	2 (2.4%)	8 (6.8%)	10 (5.0%)
**AJCC stage (8th edition)**			
Stage IIIA	12 (14.5%)	9 (7.7%)	21 (10.5%)
Stage IIIB	22 (26.5%)	30 (25.6%)	52 (26.0%)
Stage IIIC	47 (56.6%)	73 (62.4%)	120 (60.0%)
Stage IIID	2 (2.4%)	5 (4.3%)	7 (3.5%)
**BRAF status**			
Negative	38 (45.8%)	64 (54.7%)	102 (51.0%)
Positive	35 (42.2%)	39 (33.3%)	74 (37.0%)
Unknown	10 (12.0%)	14 (12.0%)	24 (12.0%)
**ECOG**			
0	67 (80.7%)	89 (76.1%)	156 (78.0%)
1	3 (3.6%)	3 (2.6%)	6 (3.0%)
Unknown	13 (15.7%)	25 (21.4%)	38 (19.0%)
**At least one documented comorbidity**			
No	46 (55.4%)	48 (41.0%)	94 (47.0%)
Yes	37 (44.6%)	69 (59.0%)	106 (53.0%)

Patient demographics and disease characteristics at start of first pembrolizumab treatment in the adjuvant setting stratified by gender. N: number of patients included in the analysis; SD: standard deviation; Min: minimum; Max: maximum; MUP: melanoma of unknown primary; AJCC: American Joint Committee on Cancer; BRAF: BRAF mutation status; ECOG: Eastern Cooperative Oncology Group.

**Table 2 cancers-16-03558-t002:** Time on treatment and reason for end of adjuvant treatment.

Time on Adjuvant Treatment	Female (N = 83)	Male (N = 117)	Total (N = 200)
Events, n (%)	69.0 (83.1%)	97.0 (82.9%)	166.0 (83.0%)
Median TOT [months] (95% CI)	11.1 (9.6–11.8)	9.9 (6.9–11.6)	11.1 (9.2–11.5)
**Time on treatment probability (95% CI) ***			
6 months	73.8 (64.7–84.1)	61.3 (52.9–71.1)	66.5 (60.2–73.6)
9 months	63.3 (53.4–74.9)	54.8 (46.2–64.9)	58.3 (51.7–65.8)
12 months	53.5 (43.3–66.0)	45.3 (36.7–55.9)	48.7 (42.0–56.6)
**Reason for end of adjuvant treatment**			
Regularly ended	32 (38.6%)	40 (34.2%)	72 (36.0%)
Disease progression	19 (22.9%)	34 (29.1%)	53 (26.5%)
Treatment ongoing	12 (14.5%)	19 (16.2%)	31 (15.5%)
Toxicity	9 (10.8%)	11 (9.4%)	20 (10.0%)
Patient’s wish	4 (4.8%)	4 (3.4%)	8 (4.0%)
Lost to follow-up	1 (1.2%)	2 (1.7%)	3 (1.5%)
Investigator’s decision	0 (0%)	1 (0.9%)	1 (0.5%)
Death	1 (1.2%)	0 (0%)	1 (0.5%)
Other	1 (1.2%)	5 (4.3%)	6 (3.0%)
Missing	4 (4.8%)	1 (0.9%)	5 (2.5%)

Time on treatment (TOT) and reason for end of adjuvant treatment stratified by gender. N: number of patients included in the analysis, CI: confidence interval. * Estimates based on Kaplan–Meier survival analysis.

**Table 3 cancers-16-03558-t003:** Time on treatment rates at 6, 9 and 12 months.

	6-Month on- Treatment Rates	9-Month on- Treatment Rates	12-Month on- Treatment Rates
**Age**			
<70 years	70.5 (63.0–78.8)	59.9 (51.9–69.1)	52.2 (44.1–61.8)
≥70 years	58.3 (47.2–71.9)	55.0 (44.0–68.9)	41.1 (30.0–56.2)
**Gender**			
Female	73.8 (64.7–84.1)	63.3 (53.4–74.9)	53.5 (43.3–66.0)
Male	61.3 (52.9–71.1)	54.8 (46.2–64.9)	45.3 (36.7–55.9)
**BRAF status**			
BRAF negative	64.0 (55.0–74.5)	59.6 (50.5–70.5)	49.9 (40.5–61.4)
BRAF positive	64.9 (54.9–76.7)	49.2 (38.9–62.3)	41.8 (31.7–55.0)
**AJCC stage (8th edition)**			
Stage IIIA	85.7 (72.0–100)	73.8 (56.3–96.8)	61.5 (42.5–89.2)
Stage IIIB	66.2 (54.3–80.6)	62.2 (50.1–77.1)	53.5 (41.2–69.5)
Stage IIIC	63.1 (54.8–72.6)	53.0 (44.6–63.1)	43.1 (34.7–53.6)
Stage IIID *	66.7 (37.9–100)	66.7 (37.9–100)	66.7 (37.9–100)

Time on treatment (TOT) at 6, 9 and 12 months. AJCC: American Joint Committee on Cancer; BRAF: BRAF mutation status. * Estimates are uncertain due to the small number of patients (n = 7).

**Table 4 cancers-16-03558-t004:** Recurrence-free survival and distant metastasis-free survival rates at 18 months.

	18-Month RFS Rates (N = 200)	18-Month DMFS Rates (N = 200)
**Age**		
<70 years	59.0 (50.3–69.3)	72.2 (63.8–81.7)
≥70 years	60.6 (48.3–76.0)	64.7 (52.5–79.9)
**Gender**		
Female	59.9 (48.8–73.6)	70.7 (60.1–83.1)
Male	59.8 (50.7–70.5)	69.6 (60.5–80.1)
**BRAF status**		
BRAF negative	59.5 (49.4–71.7)	74.0 (64.5–84.9)
BRAF positive	53.4 (42.4–67.2)	61.1 (50.0–74.6)
**AJCC stage (8th edition)**		
Stage IIIA	76.6 (55.8–100)	82.0 (65.2–100)
Stage IIIB	66.7 (53.9–82.6)	77.2 (64.8–91.9)
Stage IIIC	53.0 (43.8–64.2)	64.4 (55.1–75.2)
Stage IIID *	66.7 (37.9–100)	83.3 (58.3–100)

Recurrence-free survival (RFS) and distant metastasis-free survival (DMSF) at 18 months. N: number of patients included in the analysis; AJCC: American Joint Committee on Cancer; BRAF: BRAF mutation status. * Estimates are uncertain due to the small number of patients (n = 7).

**Table 5 cancers-16-03558-t005:** Time to next treatment and overall survival rates at 18 months.

	18-Month TTNT Rates (N = 200)	18-Month OS Rates (N = 200)
**Total [%] (95% CI)**	64.5 (57.3–72.6)	88.1 (82.7–93.8)
**Age**		
<70 years	65.6 (56.9–75.7)	90.6 (84.8–96.9)
≥70 years	62.0 (50.1–76.7)	82.3 (71.5–94.8)
**Gender**		
Female	63.5 (52.4–77.0)	90.4 (83.2–98.3)
Male	65.1 (56.1–75.6)	86.5 (79.1–94.5)
**BRAF status**		
BRAF negative	54.9 (44.0–68.6)	81.0 (70.7–92.9)
BRAF positive	66.2 (56.2–78.0)	91.4 (85.3–97.8)
**AJCC stage (8th edition)**		
Stage IIIA	80.7 (62.5–100)	100 (100–100)
Stage IIIB	72.1 (59.7–87.1)	91.2 (82.1–100)
Stage IIIC	57.7 (48.3–68.8)	85.5 (78.2–93.4)
Stage IIID *	83.3 (58.3–100) *	83.3 (58.3–100) *

Time on next treatment (TTNT) and overall survival (OS) at 18 months. N: number of patients included in the analysis, AJCC: American Joint Committee on Cancer; BRAF: BRAF mutation status. * Estimates are uncertain due to the small number of patients (n = 7).

## Data Availability

Data are contained within the article and Appendix A.

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
