# Peer review of "Adjuvant Use of Pembrolizumab for Stage III Melanoma in a Real-World Setting in Europe"

_cancers, 2024, doi:10.3390/cancers16213558_

Round 1

Reviewer 1 Report (New Reviewer)

Comments and Suggestions for Authors

In the present manuscript author try to capture novel concept of managment of melanoma. Overall manuscript is looking good and the improvment of all figure need to improve. Furthermore in the study need to identify any previously treated with other anti-cancer drugs. Also need to understand the co-morbid patient in this study . 

Author Response

Comment 1: In the present manuscript author try to capture novel concept of managment of melanoma. Overall manuscript is looking good and the improvment of all figure need to improve.

Response 1: The figures (in particular containing the Kaplan-Meier) will of course be adapted to the Journal's style in case of publication.

Comment 2: Furthermore, in the study need to identify any previously treated with other anti-cancer drugs.

Response 2: All patients were treatment naive for other anti-cancer drugs. An appropriate wording has been added.

Reviewer 2 Report (New Reviewer)

Comments and Suggestions for Authors

The authors analyzed 200 patients with stage III melanoma with lymph node involvement from Europe who had complete resection and received adjuvant treatment with pembrolizumab. The authors used conventional methods to analyze the data, and although the research results found differences between countries or regions, there was no new understanding.

The article is similar to "Five-Year Analysis of Adjuvant Pembrolizumab or Placebo in Stage III Melanoma", but the authors did not cite it. In addition, the most recent reference is only one from 2022, which also shows that the author did not refine the problem that the article aims to solve. The author needs to reconsider the research highlights of the article.

Author Response

Comment 1: The authors analyzed 200 patients with stage III melanoma with lymph node involvement from Europe who had complete resection and received adjuvant treatment with pembrolizumab. The authors used conventional methods to analyze the data, and although the research results found differences between countries or regions, there was no new understanding.

Response 1: Thank you very much for the comment. We have elaborated the particular strength of the paper addressing a more representative population than reflected by typical trial participants.

Comment 2: The article is similar to "Five-Year Analysis of Adjuvant Pembrolizumab or Placebo in Stage III Melanoma", but the authors did not cite it.

Response 2: We agree on the remarks and have referred to the long-term Keynote 54 data and also included the updated citation in the manuscript.

Comment 3: In addition, the most recent reference is only one from 2022, which also shows that the author did not refine the problem that the article aims to solve.

Response 3: We have updated the cited literature appropriately

Comment 4: The author needs to reconsider the research highlights of the article

Response 4: We have re-drafted the respective portion of the discussion and conclusion.

Round 2

Reviewer 2 Report (New Reviewer)

Comments and Suggestions for Authors

The authors revised based on the suggestions and the quality of the article improved.

This manuscript is a resubmission of an earlier submission. The following is a list of the peer review reports and author responses from that submission.

Round 1

Reviewer 1 Report

Comments and Suggestions for Authors

The quality of English needs to be checked by the Editor, a few spelling mistakes were spotted throughout the text.

- page 7, please amend the figures, centre them so the curves b and d can also be seen.

- page 6 line 201, lines 246-251 - error! reference.. please comment and amend this

- ideally, all the abbreviations should be spelled upon their first appearance in the text

- Table 3 - present the data on N of patients too

- in various clinics the control MRI/CT is conducted at different stages. For example, in East coast for some trial patients we conduct a regular MRI scans, although this practice is not common on the west coast of the country. So please specify with what frequency the radiology was done and describe more comprehensively the methodology for measuring distant mets.

- query - what was the rationale for testing those uncommon endpoints TOT and RFS?

- I would recommend to amend the discussion. it is currently repetitive with the results section. please discuss more on potential benefit of your results and how it can be compared with US-based trials.

Comments on the Quality of English Language

English language requires minor editing

Reviewer 2 Report

Comments and Suggestions for Authors

This article addresses a current topic, innovative therapies in the treatment of melanoma offering another perspective on long-term outcomes in this pathology.

First of all, it is a real achievement that you were able to create the first real multi-country study in Europe, with an updated database. It should be mentioned that the participating countries have demographic characteristics that differ in terms of skin color, exposure to UV radiation, and risk factors.

What is worth noting is the fact that distant metastasis free survival differs from the Keynote-054 study, but the group of patients chosen also differs in terms of average age and demographic characteristics. Considering that this is a real-world study, it would be important to find out after a randomized controlled study will be carried out to find out those results.

Also, another objective would be to pursue a follow-up and compare between patients treated with Pembrolizumab and patients with BRAF V600 mutation treated with BRAF-/MEK-inhibitors over a longer period, possibly at 5 years.

To sum up, this article makes a significant contribution to specialized literature, having well-structured information and addressing for the first time the European population from several countries, based on validated and up-to-date bibliographic sources

Reviewer 3 Report

Comments and Suggestions for Authors

Dear authors,

Congratulate to a well-written and interesting study on real-world data on adjuvant PD1 inhibition for melanoma stage III patients. The paper is well written and my only comments are:

1.     The PDF is not correctly formatted, e.g. (1) the right part of Figure 1 is not visible, (2) on page 6+7 a reference link seems to mess-up the text. Please revise for clarity.

2.     OS is not listed as an outcome, but is still reported in the Results section. Add OS as an endpoint, but also please provide a KM curve for OS which is currently not available in the manuscript.